# Autonomous Scene Exploration for Robotics: A Conditional Random View-Sampling and Evaluation Using a Voxel-Sorting Mechanism for Efficient Ray Casting

**DOI:** 10.3390/s20154331

**Published:** 2020-08-04

**Authors:** João Santos, Miguel Oliveira, Rafael Arrais, Germano Veiga

**Affiliations:** 1Department of Mechanical Engineering, University of Aveiro, 3810-193 Aveiro, Portugal; 2Institute of Electronics and Informatics Engineering of Aveiro, 3810-193 Aveiro, Portugal; 3INESC TEC—INESC Technology and Science, 4200-465 Porto, Portugal; rafael.l.arrais@inesctec.pt (R.A.); germano.veiga@inesctec.pt (G.V.); 4Faculty of Engineering, University of Porto, 4099-002 Porto, Portugal

**Keywords:** scene representation, autonomous exploration, next best view, ROS

## Abstract

Carrying out the task of the exploration of a scene by an autonomous robot entails a set of complex skills, such as the ability to create and update a representation of the scene, the knowledge of the regions of the scene which are yet unexplored, the ability to estimate the most efficient point of view from the perspective of an explorer agent and, finally, the ability to physically move the system to the selected Next Best View (NBV). This paper proposes an autonomous exploration system that makes use of a dual OcTree representation to encode the regions in the scene which are occupied, free, and unknown. The NBV is estimated through a discrete approach that samples and evaluates a set of view hypotheses that are created by a conditioned random process which ensures that the views have some chance of adding novel information to the scene. The algorithm uses ray-casting defined according to the characteristics of the RGB-D sensor, and a mechanism that sorts the voxels to be tested in a way that considerably speeds up the assessment. The sampled view that is estimated to provide the largest amount of novel information is selected, and the system moves to that location, where a new exploration step begins. The exploration session is terminated when there are no more unknown regions in the scene or when those that exist cannot be observed by the system. The experimental setup consisted of a robotic manipulator with an RGB-D sensor assembled on its end-effector, all managed by a Robot Operating System (ROS) based architecture. The manipulator provides movement, while the sensor collects information about the scene. Experimental results span over three test scenarios designed to evaluate the performance of the proposed system. In particular, the exploration performance of the proposed system is compared against that of human subjects. Results show that the proposed approach is able to carry out the exploration of a scene, even when it starts from scratch, building up knowledge as the exploration progresses. Furthermore, in these experiments, the system was able to complete the exploration of the scene in less time when compared to human subjects.

## 1. Introduction

The emergence of intelligent autonomous systems in key societal sectors is thrusting innovation actions and the materializing of technological novelties in the domain of autonomous spatial exploration. The contemporary self-driving cars uprising, the mass use of unmanned aerial vehicles, and the ongoing fourth industrial revolution are clear examples of scientific and commercial drivers for this particular domain of research. We consider as an example of the potential value of the application of smart autonomous exploration techniques the case of the often refereed Industry 4.0 movement [1], where Cyber-Physical Production Systems (CPPS) such as autonomous robotic systems are helping companies to cope with acute alterations in production and business paradigms [2], frequently needing to operate in highly unstructured and dynamic environments [3]. As a direct result, spatial exploration and object recognition by robotic systems is a key part of many industry-wide applications and services. Despite being increasingly efficient, this recognition is often based on the processing of information collected at a single moment, such as images or point clouds. Thus, most implementations assume that the object recognition process is static, rather than dynamic. To make this process dynamic, a system needs to have some kind of exploration abilities, i.e., being able to search and discover information about a given goal.

The increasingly more dynamic and smart implementations will force a paradigm shift in the field of exploration by intelligent systems, since these give a new level of adaptability to novel automation tasks that, at this point, are not possible. These implementations can range from smarter bin-picking, reconstructing the scene at each iteration for better adaptation, to a system that collaborates with humans, detecting when something has been repositioned or removed from its working space, and reacting accordingly. For this shift to be feasible, the new solutions need to be at least as reliable, fast, and secure as is already standard in today’s industry.

The ability to explore a new environment, quickly and efficiently, developed to increase our survival skills, is intrinsic to humans [4]. Our eyes evolved to catch the tiniest of details and our brains got smarter in understanding the environment and processing it to evaluate the goal, extrapolate the information gathered, and choose where to move next. This raises the questions: How to give a piece of hardware that cannot think for itself the human abilities to know what it has and has not already seen? How to know where it should observe the scene next to gather the newest information available?

Exploration is the process by which novel information is acquired. In the context of spatial exploration, the information that is gathered concerns the state of occupancy of the several portions of the scene. Thus, the goal of scene exploration is to measure the state of occupancy of the maximum possible regions of the scene. An exploration system requires the ability to conduct active perception, i.e., to deliberately change the point of view of the sensor with which it observes the scene. An intelligent exploration system should analyze the current knowledge it has about the scene and, based on this information, estimate the next point of view to which it should move. This is what we refer to as NBV. The NBV is selected based (most often solely) on the amount of information it is estimated it will provide. This exploration step is then repeated until a termination criterion is met. Common termination criteria are the mission time, the knowledge about a scene, the lack of progress in the last set of exploration steps, among others.

As described above, the explorer system must contain a representation of the occupancy of the different portions of the scene. Furthermore, this representation must be adaptive, in the sense that new sensor measurements must be integrated into the existing representation. With such a representation it is then possible to address the problem of selecting the next best view. However, taken at face value, the problem of finding the best view is a cumbersome one because the set of possible next views is infinite. The solution for this is to create a finite set of views, where each is tested to estimate the information it should generate if selected. These views are often randomly generated. The number of possible views combined with the complexity of the test conducted for each view will define the computational efficiency of the approach. Within preset time boundaries, the goal is to assess as many views as possible. One way to address this problem is to create efficient yet accurate assessment methodologies. Another is to make sure the views which are generated are feasible and plausible. Feasible because they are within reach of the explorer system, and plausible because they are aimed at (observing) the volume that is to be explored. This is done with a conditioned randomized procedure. Without it, it would be likely that there are several of the generated views that would be unreachable by the system, or even reachable but not looking at the region of interest.

All these issues are tackled in this paper. As such, our contributions are the following:To make use of an efficient and updatable scene representation, i.e., OctoMap;To design a generator of plausible and feasible possible viewpoints, which increases the quality of the viewpoints that are tested, thus improving the quality of the exploration;To proposed a viewpoint evaluation methodology which sorts the voxels for which rays are cast, which enables the algorithm to skip a great number of tests and run very efficiently, thus allowing for more viewpoints to be tested for the same available time;To provide extensive results in which the automatic exploration system is categorized in terms of the efficiency of exploration, and compared against human explorers.

The developed method was used to reconstruct a couple of scenarios to assess repeatability. Also, it was tested against humans, using the part of the tools created during the development of the autonomous system, to infer if it was advantageous or not to use this exploration methodology. Furthermore, an industrial prototype application on the scope of a European-funded research initiative was considering, attesting to the potential applicability of this solution to address some of the aforementioned contemporary industrial challenges in the domain of autonomous exploration.

The remainder of this document is divided into four additional sections: In Section 2, a theoretical overview of some of the concepts addressed by this work as well as several comparable approaches are presented, emphasizing their strengths and shortcomings. Section 3 presents the proposed approach, detailing the methodology, roadblocks, and solutions developed to accomplish the enlisted objectives of this work. This includes the manipulator’s configuration to accept external commands, the mounting and calibration of the camera, and the exploration algorithms. In Section 4, the authors introduce two case studies and an industrial prototype application where the developed work is tested, following a comparison between the autonomous system and several humans’ abilities to explore a given scene. Still in this section, plausible implementations and further improvements are discussed. Finally, in Section 5, conclusions are outlined and future work topics are discussed.

## 2. State of the Art

### 2.1. Spatial Data Representation

A fundamental step in empowering autonomous systems to operate in unstructured and dynamic environments is to gather information to allow evaluation of their surrounding environment. Point clouds are one of the most common forms of 3D data acquisition. Although they can be easily acquired, with low-cost sensors that have decent performance, especially at close range [5], this raw data can easily become hard to process due to the sheer amount of information when considering large environments. To solve this drawback various methods exist. What these methods try to accomplish is merging some of the points in the cloud, losing as little information as possible.

One approach is to create a 2D grid representation of the surroundings, where each cell also contains a value, typically the possibility of the grid being occupied [6] or height [7]. This last form of 3D representation is referenced as elevation maps. Elevation maps can be classified as 2.5D models because they have the 2D position information of each cell, which in turn contains a corresponding height value. This height value is obtainable by various methods. One of the simplest is averaging the height coordinate of all points falling inside a given cell. The biggest advantage is the filtering of outlier measurements, but this also leads to the inability of capturing overhanging structures (like bridges, for example), according to [8].

To overcome this inability [9] developed the extended elevation map. The indicated approach can find gaps by analyzing the variance of the height of all measurements in a given cell. If this variance exceeds a defined threshold, the algorithm checks if this set of points contains a gap that is higher than the robot they used for testing. If so, the area under the overhanging structure can be properly represented by ignoring the points above the lowest surface.

Although the extended elevation maps bring improvements to this type of representation, its applicability is limited to applications requesting only terrain-like representation. This is a major issue in the scope of this work since a truly 3D representation is needed. For example, if the environment that we want to explore only contains a table, using standard elevation maps would give a fully occupied volume below the table. On the other hand, an extended elevation map would not properly represent the tabletop, not treating it as occupied.

That being said, some other methods also use the same philosophy of dividing the space into a 3D grid, called voxel grids [10]. A voxel grid is generated when the scene is divided into several, recursively smaller, cubic volumes, usually called voxels (hence the name voxel grid). This division translates the continuous nature of the world to a discrete interpretation. To represent the scene, each voxel can have one of three different states: free, occupied, or unknown. In a newly generated voxel grid, all cells (i.e., voxels) start as unknown, since no information has been gathered yet. When a point cloud is received, ray-casting from the camera pose to the data points is performed. Ray-casting consists of connecting the camera position and a point of the cloud with a straight line [11]. Then, in all voxels that are passed through by a ray we assume that they do not contain an obstacle (i.e., there is not the acquisition of a point) and so, are identified as free. If, in contrast, a point is sensed inside a voxel, it is considered to be occupied. This is the core of how states are defined but, in practice, for a voxel to be considered occupied it must contain more than a given number of points. The opposite needs to be true for a cell to be free.

By expanding the concept of grid to a volumetric representation, voxel grids can, as accurate as its resolution (the size of a voxel side), reconstruct overhanging structures and virtually any object. Yet they are not a feasible solution as they lack a fundamental characteristic needed for this work, updatability. In voxel grids, once a voxel state is defined, it cannot be changed. This prevents the addition of new information (for example, by viewing by a different pose) to the reconstruction. Furthermore, the resolution is fixed, compromising efficiency, especially in cases that need a large voxel grid with fine resolution.

Yet, even with these drawbacks, voxel grids lay the foundations for OcTrees [12], which deal with multiple resolutions within the same volumetric representation. Building upon the voxel grid methodology, OcTrees are a data structure that divides the pretended portion of the world into voxels but allows those voxels to have different sizes. In an OcTree, each voxel can be divided into eight smaller ones, until the minimum set resolution is reached [13]. It is this subdividing property of the voxels that establishes a recursive hierarchy between them [14]. This resolution multiplicity means that some portions of the space that have equal state (this is, are free or occupied) can be agglomerated in one larger voxel. This is called pruning. Pruning occurs when all eight leaves (the nodes of the OcTree that have no children) of a node are of the same type, so they can be cut off, requesting lesser memory and resources, which improves efficiency.

Recursing through the OcTree structure allows most of the information of the inner nodes to be obtained. For example, the depth of a node can be calculated by adding one to its parent depth. A similar thought can be traced to the position and size of a node. To optimize memory efficiency, any information that is computable by traversing the tree can be omitted [15]. Furthermore, in cases where the mapping is performed for large spaces and high resolutions are required, voxel interpolation methods can be applied to obtain similar results with lower memory consumption and better framerates [16].

Although the efficiency issue of voxel grids is solved, OcTree still lacks updatability that by itself, is already a challenge, requesting a criterion which leads to a voxel changing its state. In a voxel grid, each voxel can store a value that defines if it is free or occupied. In the case where that measurement was not taken, the voxel encounters itself in an unknown state. In a probabilistic voxel space, its value ends up being the probability of a voxel being occupied. Since we are dealing with probabilities, a new measure of the environment can be integrated to update those probabilities. With a changing occupancy value comes the possibility of a voxel changing its state. There are several ways in which the integration of a new measure can occur.

OctoMap [17] manages this process using the log-odds of a voxel being occupied instead of the direct probability value. This causes the substitution of multiplications by sums, achieving a more efficient algorithm. In the works of [18], the OctoMap voxel space update procedure is improved in a way that now considers reflections. Using the MURIEL method [19], this approach adds a parcel to the overall probability of a voxel being occupied that is referent to measures in specular surfaces. In addition to this change, ref. [18] also uses Dynamic Multiple-OcTree Discretionary Data Space. This structure allows the use of multiple octrees for discretizing the space.

OctoMap delivers an efficient data structure based on OcTrees that enables multi-resolution mapping, which can be exploited by the exploration algorithms [20]. Using probabilistic occupancy estimation, this approach can represent volumetric models that include free and unknown areas, which are not strictly set. As proven by [18], OctoMap still has room for improvements such as being able to consider reflections and to be based on the actual sensor model. Yet it is the most efficient and reliable 3D reconstruction solution currently available.

### 2.2. Intelligent Spatial Data Acquisition

With a found way to accurately represent the environment and its changes, the next logical step is to feed the map with measurements. These measurements could be taken by putting the object/scenario on a rotating platform [21] or by moving the camera through a set of fixed waypoints. Neither of these solutions is very good in terms of adaptability to the environment, being useful only in a few situations.

An intelligent system needs to adapt to anything that is presented to it, deciding by itself to where it should move the camera, to obtain the maximum possible information about the scene. In some particular cases, part of the scene cannot be observed, for example, a box with all six sides closed. The system must understand when it has found a situation like this and give the reconstruction process as completed.

In [22] the goal was to find a sequence of views that lead to the minimum time until the object was found. To achieve this, they determine the ratio between the probability to completely find the target—with a given pose—and the transition time that takes to get there. With their utility function, the planner favored three situations: (i) locations that are closer to the previous position, (ii) locations where the probability of finding the object is very high or (iii) a combination of the previous two points. The innovation in this approach is that they plan several steps and, after each pose evaluation, the list of poses that are intended to be visited is updated. By using, as the authors describe, “the probability to find a target with observation after having already seen the sequence they assume that all possible positions are determined by sampling, and so the problem is not only to generate them but also to choose which ones to visit and in what sequence” [23].

Since replanning occurs in every pose, a condition to stop the planning procedure must exist. In [22] the planning ends when the probability of finding the unknown voxels, by adding a new pose, falls below a user-given threshold. The described methodology worked very well on the authors’ experiments. Yet, taking into account that this work will use a robotic manipulator (that compared with the PR2 robot is not able to move as freely) the moving time component used in their approach will not reveal a great significance.

When [24] tackled the problem of semantic understanding of a kitchen’s environment, they used a 2D projection of the fringe voxels on the ground plane allied to the entropy of the space within the view frustum (i.e., the region of space that is within the Field of View (FOV) of the camera). This entropy is the measurement of how many voxels in different states are possibly detectable by the pose being evaluated. The more different they are, the higher the entropy will be. On the other hand, fringe voxels are considered windows to the unexplored space because they are marked as free but have unknown neighbors.

First, it is necessary to introduce the concept of a cost map. A cost map is a data structure able to represent places in a grid where it is or is not safe for a robot to move in. What [24] does is to create two separate cost maps for the fringe ($C_{F}$) and occupied ($C_{O}$) voxels and then combine those values. The combination makes each cell store the minimum value between $C_{F}$ and $C_{O}$ (representing the lowest risk of collision). Now, choosing the maximum value from the combined cost map they achieve a pose from which as many fringes and occupied voxels are observed, with little risk of collision. It is necessary to observe several occupied voxels because there is a need to achieve a 50% overlap with this new view and the already created map. In [24] moving expenses were not considered, which meets the intended for this work, but further adds a new type of voxel (fringe), which makes sense in the large rooms they focused and, furthermore, removes the need to evaluate occlusions. For object exploration, these fringe voxels increase the complexity of the system in a direction that will not produce any significant advantage. We need, in each new pose, to evaluate the largest possible number of unknown voxels, whether they are close to free space or not.

An alternative is to estimate the score of a pose by the volume it could reveal. As with what [24] called fringe voxels, ref. [25] define frontier cells as the ones that are known to be free and are neighboring any unknown cell. Their goal is to remove all unexplored volume, called voids. For each frontier cell, a set of utility vectors are generated. These vectors contain information about the position of this cell, the direction from the center of the void to the cell, and the expected score value from that cell (i.e., the total volume of the cells that are intercepted with ray tracing, from the center of the frontier cell to the center of the void). For each free cell that is intersected by a utility vector, ray tracing is performed with its direction, and the utility score stored. Of all these viewpoints, those who are outside the robot’s workspace are pruned and the remaining are sorted by their util(c) values for computing valid sensor configurations. In other words, this means—after sorting the viewpoints by their score—sampling some valid camera orientations and perform, again, ray tracing. After a given number of poses has been computed, the next best sensor configuration is the one that has the highest utility score (i.e., allows the observation of most volume). The recognition process terminates when the last view utility is null.

Another possibility to choose between the sampled poses is not by their total information gain but, instead, ranking them compared to each other. With this goal, ref. [26] proposes a utility function that compresses the information gain and robot movement costs of a given view with the cumulative information gains and movement costs of all view candidates. When a pose gives the maximum relative information gain with the minimum relative cost, the authors consider that they achieved the NBV. When the highest expected information gain of a view falls below a defined threshold the reconstruction is completed. Until this point, the utility functions are composed of a small number of inputs (one to two), but larger equations are possible.

### 2.3. 3D Object Reconstruction and Scene Exploration

Trying to get an algorithm that was optimal for motion planning in 3D object reconstruction, ref. [27] came up with the goal to maximize a utility function composed by four distinct inputs: (i) position, related to the collision detection, (ii) registration, a measurement of overlap between views, (iii) surface, which evaluates the newly discovered surface, and (iv) distance, to penalize the longer paths.

If a position requires a path that enters in collision with the environment, the position portion of the equation is set to zero, eliminating that candidate. The registration part is evaluated as one if the overlap portion of the new view is above a given threshold, or zero otherwise. Next comes the first non-binary value, related to the new surface measured, that gives the number of unknown voxels sensed with that view, normalized by the amount of the total unknown voxels in the workspace. The final value is inversely proportional to both translation and orientation movements since the goal is to maximize, the smaller the length, the greater this value will be. To stop the reconstruction process, a condition is used which is based on the number of sensing operations needed to have a certain confidence of having sensed a portion of the object.

This creates two scenarios, the first in which the percentage of voxels inside the object is known a priori and the number of sensing operations is calculated once, or the second one where that value is unknown and so, this number is recalculated in every cycle. When the sensed portion does not change within this amount of operations, the task is considered to be over with the requested certainty.

One thing that has been missing in all the described utility functions is a parcel specifically dedicated to the modelling part of the problem. An introduction to this methodology is performed by [28], in which two goals exist, the full exploration of the scene and the modelling of it with decent surface quality. These two goals gain or lose relevance in the output of the utility function based on the number of scans performed, making the best NBV the one which gives the most entropy reduction with the most surface quality. In their respective setups, both [29] and [28] use an RGB-D camera combined with a laser striper. Allied, these devices combine the faster and completer overview of the scene provided by the RGD-D camera with the superior surface quality reconstruction from the laser striper. In the latter case, initially it is wanted a rough but broad mesh of the object but, later, the quality of that mesh begins to matter the most. To perform this change, a transition value is chosen. This value comes in the form of a weight, based on the number of scans already performed of the scene.

In [28] the stopping criteria were more recognition oriented, so the process was established to end when, at least, 75% of the object was modelled. Yet, the end of the process can be reached when there is already enough mesh coverage and relative point density. If neither is reached, ref. [18] also implement a maximum number of scans criteria. They also introduce a method for switching from generating candidate poses to rescanning (in their case, rescanning holes in the mesh) if the increase in coverage of one scan compared to its previous is less than 1%.

All the methods already described lay in predefined evaluation functions and human set threshold values. These methods are considered to be a more classical approach to solve a computational task. Currently, the trend to solve such tasks is to use neural networks and provide systems with artificial intelligence. These systems can be taught by analyzing examples, like manually labelled data. With the spreading of neural networks and artificial intelligence, the usage of this new and powerful methods has already been accomplished in some particular situations.

Remembering that each voxel stores a probability value of being occupied, ref. [30] uses a utility score function that defines the decrease in uncertainty of surface voxels (voxels that intersect the surface of the object being reconstructed) when a new measurement is added. The goal here is to predict this value without accessing the ground truth map. Using a ConvNet architecture, trained with photo-realistic scenes from Epic Game’s Unreal game engine, the 3D scene exploration occurs in episodes. An episode starts by choosing a random starting position to mark it as free space. Progressively, at each time step, the set of potential viewpoints is expanded to the neighbors of the current position, all of them are evaluated and the robot moves to the best one. The authors claim that their model can explore new scenes other than the training data but is dependent on the voxel distribution of the said data. They also do not contemplate moving objects in the scene and are bound by the resolutions and mapping parameters of the training data.

### 2.4. Contributions beyond the State of the Art

The proposed approach for solving the problem planned for this work consists of an adaptation and blend between [25] view-sampling method, ref. [26] utility function and [27] surface parcel computation. With a set of points representative of where the camera should look seems a good starting point for the view-sampling procedure, increasing the chance of those pose being plausible. To be plausible, a pose must have, somewhere in their FOV, at least one unknown voxel expected to be evaluated. Comparing and ranking those poses against each other makes it possible to introduce stopping criteria that directly correlates to the first NBV and so, it is expected that in each exploration iteration (i.e., the movement of the robot to a new viewing pose) the NBV score always decreases. The use of neural networks was not considered since their potential in this field has not yet been proved to introduce advantages that overcome the limitations and, maybe even more relevant, the time-consuming process of training and evaluating it.

For the reason that the proposed approach relies on an active perception system [31], after deciding where the camera should move, a path to get there must be planed. This path cannot enter in collision with the environment, under the penalty of changing it or causing damage to the hardware. This means that to correctly plan the path of every element of the autonomous system (in the case of this work, the 3D camera and the robotic manipulator), their relative positions must be known. Eventually, we could also take advantage of this path to enforce on it a subset of goals poses that would increase even further the amount of collected information [32].

## 3. Proposed Approach

The ability to conduct autonomous exploration requires an accurately calibrated robotic system so that the system can anticipate the pose of the in-hand sensor as a function of the manipulator’s joint configuration. Also, a criterion for the adequate selection of the NBV is required, as a means for the system to plan the most adequate motion to explore the scene. In this sense, the NBV is defined as the camera pose which maximizes the amount of new information gathered about the scene, which is dependent on the current state of the reconstruction of the environment.

To address the first requirement, a calibrated system, a hand-in-eye calibration war performed. This procedure returns the geometric transformation between the manipulator’s end-effector link and the camera sensor link and works by capturing multiple and representative views of a calibration pattern. The ARUCO / VISP Hand-Eye Calibration (https://github.com/jhu-lcsr/aruco_hand_eye (accessed on 16 March 2019)) provides an easy way to implement Aruco-based extrinsic calibration to the system.

Providing a way to explore a given volume autonomously is the overall objective of this work. To do so, the architecture depicted in Figure 1 is proposed. This architecture allows for a fully autonomous exploration that always guarantees the manipulator’s movement to the pose estimated to give the most information gain and which it can plan for. The next subsections will present in detail the functionalities described in the proposed architecture.

### 3.1. Defining the Exploration Volume

The objective of the exploration may not be to explore the entire room but, most often, just a small portion of it. In addition to this, it is intended that these exploration systems operate online, and a delimitation of the volume to be explored is an effective way of speeding up the process. To address this, we propose the concept of exploration volume, which is intended to be explored. In other words, the system considers that there is no information gain in observations outside this volume.

Notice that this cannot be done just by discarding the points outside the exploration volume, because the construction of an OcTree map builds not only from the points themselves but also from the ray that goes from the sensor to that point. A point that lies outside the exploration volume may still create a ray that is important to build the representation inside the exploration volume. The solution is to define a second, larger region named point cloud filter volume. In these experiments, a region five times larger than the exploration volume was used. This value was empirically obtained as the best compromise between memory usage and the performance of the map reconstruction. Points outside this larger region are then discarded. Figure 2 shows an example which explains the purpose of the point cloud filter volume. Since in Figure 2a the exploration and filter lines are coincident, the beam bounces back outside the filter, disabling its registration, making it impossible to mark the black voxel as free. On the other hand, in Figure 2b, the point is detected since it is inside the point cloud filter, allowing the unveiling of the black voxel free state.

This solution ensures two things: the OctoMap does not become very large and detailed in unimportant regions and that there is not an excessive amount of information filtered out. Figure 3 displays the complete filtering procedure.

### 3.2. Finding Unknown Space

Having developed an effective way of constructing online an OcTree representation of a volume to be explored, the goal now is to assess which regions in that volume have not yet been explored. Notice that ocTrees do not store information about unknown voxels explicitly. To extract this information, we iterate through all voxels of the OcTree and verify if their state remains undefined.

It is also important to group the set of unknown voxels into spatially connected clusters. The reason for having these clusters of unknown space is to be able to generate plausible sensor poses as is detailed in Section 3.3. Clusters are built by iterating through all the centers of the unknown voxels, and testing if the distance to the neighbors is within a predefined distance (in this case, just over the double of the OctoMap’s resolution). An iterative procedure forms clusters that grow until all the neighbor voxel centers are too distant. The procedure is similar to a flood fill or watershed algorithm in image processing. The result of the clustering procedure can be observed in Figure 4a. For each cluster of unknown voxels we compute the center of mass of its constituent voxels: see Figure 4b for the centroids that correspond to the clusters of Figure 4a.

### 3.3. View-Sampling

View-sampling is the process by which several views, i.e., camera poses, are generated. Notice that the goal is to generate a set of random camera poses and to select the best one from that set. In this context, the best view would be the one that maximizes the information gain. To select the best view, each pose must be evaluated. This evaluation process is time-consuming, which in turn restricts the number of evaluated views per exploration loop.

Given this, it is nonsensical to generate (and evaluate) entirely random views, since a significant number of these would not be able to gather information about the exploration volume. On the other hand, it is important to maintain an ingredient of randomness in the generation of the views, so that it is virtually possible to visit any view. To address this, we propose a procedure to which we refer to as conditional randomized view-sampling. Each view is a sensor pose in world coordinates. The process divides the views into two components: position and orientation.

The position components are randomly sampled within the reach of the robotic manipulator using MoveIt [33]. To prevent poses where the manipulator itself occluded the exploration volume, the position generation is also limited to the front hemisphere of the working space. This first stage generates a set of incomplete sensor poses that contain only the translation components. For reaching those poses, an orientation component is generated. The center of observation, i.e., the center of the unknown voxels cluster to which each sensor pose is associated with, is used to generate the orientation. Using the position of the pose and the center of observation a *z* versor (the direction which the camera will look at) is computed coincident to the line that unites both points. The *y* versor comes from the cross product between the *z* versor and a random vector and the *x* versor is also the cross product between the already defined *z* and *y* versors. Figure 5 gives an example of a set of generated poses following this procedure.

This method gives a solution to produce a set of plausible poses. To decide which to visit we need to score and rank them all. The first request to give a score to a pose is knowing how much unknown voxels it unveils. This procedure takes into account the reach of the robotic system, i.e., if the system can position the camera at the set of plausible poses.

View-sampling requires information about the center of the clusters, so they acquire the best orientation. The total number of poses is a parameter and the algorithm evenly distributes that quantity by all the found clusters, as in Equation (Equation 1),
(1)$$ppc=\mathrm{round}\left(\frac{N_{poses}}{N_{clusters}}\right),$$
where ppc is the number of poses per cluster, $N_{poses}$ and $N_{clusters}$ are, respectively, the number of total desired poses and found clusters. The advantage of this method is that the total number of poses is independent of the amount of clusters found, allowing for a more stable time consumption per exploration iteration, when compared to if the amount of poses was set for each cluster found.

### 3.4. Generated Poses Evaluation

The next step is to be able to evaluate each of the previously generated poses. The evaluation of these poses is based on the estimated amount of volume that is (or can be) observed by the robot when positioned on that pose. As noted before, we are simply estimating the volume that may be discovered by a new pose. This is an estimate since it is not possible to be sure if some regions of the yet to be seen volume will be occluded.

To estimate the volume that may be observed, we propose an algorithm based on ray-casting driven by the centers of the unknown voxels that lie inside the camera frustum. Figure 5 also shows an example evaluation of a pose where the voxels expected to be known, if the robot is placed on that pose, are colored blue. A method to retrieve this information is introduced in the following subsection.

### 3.5. Voxel-Based Ray-Casting

This method starts the ray-casting on the unknown voxels instead of on the pose. This not only has the potential to give higher resolution to locals with more unknown voxels but, also, significantly decreases the number of rays to be cast. If we start the casting from the furthest voxel to the closest (relative to the pose to be evaluated) there is a high chance that the first rays intersect a large number of voxels. Assuming that in this evaluation, a voxel only needs to be visited once for its state to be defined, there is no need to cast a ray again for those voxels (that have already been intercepted), eliminating them from the queue, reducing the overall rays to be cast.

After computing the central point of all unknown voxels, we start by defining the set of those that lay within the frustum, since only these have the potential to be discovered. These points, and their corresponding voxels, are sorted by distance to the pose being evaluated. Starting from the furthest voxel, the direction from the pose to its central point is computed. This direction is used to cast a ray that if there is not any occlusion, will pass through the voxel. All the voxels that are intercepted by this ray will not be visited again, since we already have the confidence that at least one ray will hit them. In the case where the ray ends in an occupied voxel, it is necessary to take into account the voxels that are occluded. Therefore, following the same direction, from this occupied voxel until the end of the frustum, those voxels that would be intercepted by this hypothetical ray will not be visited any time and neither be assumed to be discovered. Then, for the next furthest voxel that has not been visited yet, the process repeats itself until all the volume is evaluated. Algorithm 1 describes this process.

Figure 6 represents the visual outcome of this algorithm, where is visible the casted rays (gray lines), the occupied (red), free (green), unknown (orange) and expected to be known (blue) voxels, as well the camera frustum.
**Algorithm 1:** Voxel-Based Ray-Casting.
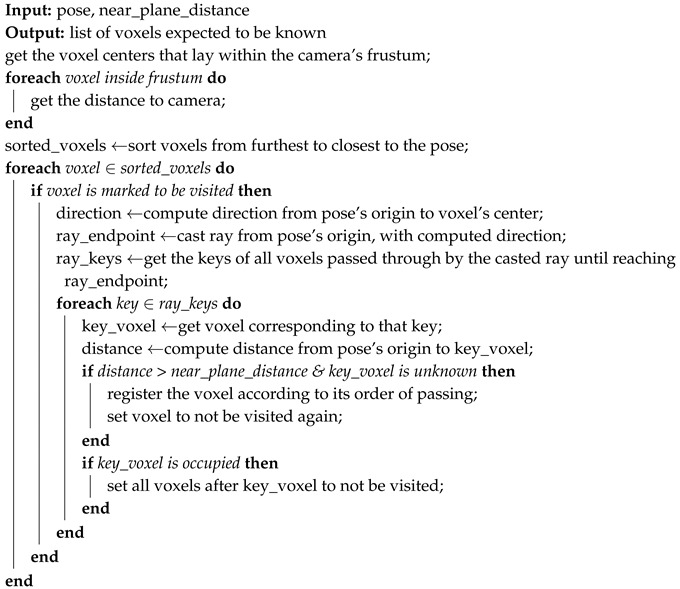


Now we want to rank the poses against each other. The metric implemented to do so, i.e., the formulation that gives a score to a pose, is based on this assessed unknown volume to be discovered.

### 3.6. Pose Scoring

To rank all the poses, a term of comparison must be set. In this work, when estimating how many voxels a pose will evaluate, we expressed this value as a volume. Then, it becomes straightforward that the term of comparison should also be a volume. As referred in Algorithm 1, the order in which the voxels are intersected is a useful information. The first voxel which is intersected is always assessed. As other voxels which lie behind the first (along the casted ray) will be visible or not depending on whether an occupied voxel surges in front of it.

Since the quantities of voxels that are the first to be intercepted by a given ray, $n_{first}$, and that are intercepted by a ray but have an unknown voxel between them and the evaluated pose, $n_{posterior}$, are known, as well as the OctoMap’s resolution, a weight w∈[0,1] can be set to give less relevance to the voxels which are uncertain to actually be observed,
(2)$$score=\frac{\left(n_{first}+n_{posterior}\times w\right)\times resolution^{3}}{volume_{outer}+volume_{inner}\times w}$$
where $volume_{outer}$ is the aggregated volume of all voxels that have at least one face exposed, and the $volume_{inner}$ is the volume of all voxels that have all their faces connected to another unknown voxel. Both volumes are computed before the exploration begins and have the objective of normalizing the pose’s revealed volume similarly as estimated.

Scoring a pose is also useful in the sense that provides stopping criteria for the process. When a set of poses are sampled and the best of their scores does not get above a given threshold (i.e., $\max\left(score_{k}\right)<threshold,k\in N_{n_{poses}}$, where $n_{poses}$ is the defined maximum number of poses to be sampled), that pose is still visited since it was evaluated but will be the last given that from this pose onward, the information gain will surely be below the pretended value. This can occur by two factors: either all the unknown volume has actually become known or the voxels that remain unknown are in unreachable places. Having gathered the desired, possible, information about the previously unknown scene, the process finishes successfully.

## 4. Results

This chapter presents results of the developed system and, when opportune, discuss its limitations, applications and possible improvements in a future continuation. The first topic to discuss is the functionality and autonomous capabilities of the developed system and for that three test scenarios will be introduced. These scenarios intend to show real-world applications. In the first two scenarios were at Laboratório de Automação e Robótica—University of Aveiro (LAR-UA) built to challenge the system, posing two different challenges. In the third scenario, we demonstrate a prototype of an industrial application scenario where this autonomous exploration system would allow the unveiling of parts inside containers for further manipulation in bin-picking operations. The industrial scenario was considered on the scope of the H2020 FASTEN R&D project [34] and required an alteration on the hardware architecture employed.

### 4.1. Test Scenario 1: Shelf and Cabinet

This first scenario consisted of a cabinet that has a shelf behind it. The shelf has some objects on the shelves to generate occlusions, and, on the cabinet, one of the drawers is semi-open, and contains an object inside, creating a volume that cannot be seen and other that can only be visualized from a narrow set of viewpoints. The complete scenario can be seen in the right column of Figure 7. Figure 7 describes the exploration process for this scenario, taking six iterations (only three shown) and sampling 150 poses for each, with the finest map resolution defined at 40 mm. The exploration volume is 3.238
m${}^{3}$ and the reconstruction ended when the NBV had a score of less than 0.1%. For this reconstruction was defined that the finest resolution was 25 mm since it gave the most details without a big compromise in performance. This resolution allows the capture of finer details.

In the first iteration (before the movement of the robot, Figure 7a) a large amount of volume is expected to be observed, due to no information about the scene. In Figure 7b, it is discernible the exclusion of some voxels in the expected volume to be known, as a result of the occlusion created by the traffic cone. Yet, after the pose was reached, we can see that the refereed voxels have in fact been evaluated (in this case, as free) in conjunction with some other in front of the cabinet. This happens because of the path taken by the camera. While moving, the camera is continuously sensing what is inside its FOV, sometimes providing information where it was not expected to be collected, depending on the path travelled. This factor was not taken into account during the evaluation of poses because it would request a great amount of computational power to evaluate all steps of the path (these steps can add up, surpassing the hundreds).

This iteration also demonstrates a characteristic movement of the robot, going from right to left to access the state of the voxels positioned where the camera has not looked up until this moment. This NBV was oriented to look towards the bigger cluster in the front right of the exploration volume. Nevertheless, some voxels belonging to different clusters are also expected to be visible and within the FOV, which increases the score of that pose. Because some were behind a solid obstacle (yet unmapped) not all of them were actually observed. In the next few iterations, the robot bounces right and left gaining information about a decreasingly unknown volume until the stop criteria are met. At the end of the process is visible the almost absence of unknown volumes (Figure 7c), which has been evaluated to be free or occupied.

The shelf and cabinet scenario provided a challenging reconstruction because of its high complexity, with multiple objects, intricate spaces, and volumes impossible to be seen, requesting from the robot several demanding poses to achieve its goal. Despite this, it took an average of eight poses to fulfil the task. The goal for this section was to provide a more qualitative evaluation and demonstrate how the process unfolds, with the quantitative results being discussed in Section 4.3. This philosophy will be maintained in this next section, where a new scenario is presented.

### 4.2. Test Scenario 2: Occluded Chair

The previous scenario lacked a big occlusion that would challenge the pose evaluation procedure, testing its real aptitude to deal with these kinds of challenges. Going even further, the system must be able to react to an occlusion that is expected to happen and move around it, visiting poses that are expected to return the biggest possible information, without interacting with the scene, i.e., without colliding with it.

This new scenario is presented in the right column of Figure 8, where the chair is visibly occluded by a tall obstacle with a triangular base. Being this tall forces the robot to look from either side, disfavoring central positions, while making it impossible—given the reach of the manipulator—to look inside of said obstacle by its only entry, at the top. The obstacle is also close enough for the robot to collide with it if no collision checking would be performed.

Two big, unknown, volumes were not sensed in this exploration: behind the chair and inside the obstacle. The unknown cluster in the rear of the chair exists solely due to the robot’s inability to reach a pose in which the camera can perceive that volume. Similarly, as already referred, the set of poses able to unveil the cluster inside the barrier would be in a very high position looking down. However, even these would not grant a complete evaluation of the said volume since the obstacle is taller than the FOV of the camera. Therefore, to completely unveil this volume, the robot had to reach a high position and then, probably, go inside it in a second one. Unfortunately, these poses and movements are out of the manipulator’s reach.

Another aspect is the poor reconstruction of the back portion of the obstacle, once again explained by the inclination of the planes. As represented in the previous figure, some voxels, in the same vertical line, are occupied and others free, when was expected for all of them to be in the same state. The most plausible explanation lays in the orientation of the walls. Given the reachability of the manipulator, most of the rays intersecting these voxels would be almost parallel to the barrier, passing through them without actually hitting the barrier, wrongly setting them as free. Yet, in another pose, possibly with a higher rotation, these rays actually bounce on the barrier, setting those voxels as occupied. This uncertainty would explain the high entropy of the reconstruction of those two back barriers.

As discussed when analyzing the reconstruction of the shelf and cabinet scenario, the first NBV does not contemplate information about occlusions. By chance, in this first iteration (Figure 8a), only the lower portion of the set is evaluated, requesting a pose capable of sensing the higher portion, which happens next. After the second exploration iteration, the barrier has already been mapped, allowing for full occlusion avoidance poses, exactly as happens in Figure 8b, where the robot, almost all stretched, looks from right to left to reveal the remaining top portion of the exploration volume. Finally, like in the previous scenario, there is the last iteration where a small number of unknown voxels are measured to fulfil the end criteria.

One of the goals when testing with this different scenario was to evaluate how the robot reacted, and planned, to an object within its reach. Figure 9 illustrates an example case where this happens. After having reconstructed part of the scene, the robot selected a pose that requested it to move across the scene but, if taken a straight path, would collide with an object it already knows the existence of. To avoid it, the camera curves down and inwards in the process, distancing itself from the barrier. When reached a position where a collision was not expected to occur anymore, an upward path was taken to achieve the goal pose. To achieve this behavior, various path planners are available and in this work, the one chosen was RRT-Connect [35] since it proved to be the one who allied efficiency, simplicity and the better interaction with the environment.

Similar to what happened for occlusions, this behavior tends to react better to the obstacles as more and more knowledge about the environment is gathered. It also gives the insurance of a completely autonomous exploration by the robot. The ability to generate a model of a new environment, without changing it with a collision is there, achieving the goal of this work. Accessing if it is advantageous to use in replacement of a human controlling the manipulator is discussed further ahead in Section 4.5.

### 4.3. Experimental Analysis of the Autonomous Exploration Performance

In this section, a set of quantitative results is provided. These results compare the technique adaptation of the algorithm to both scenarios already described. The summary of the physical characteristics and performance of the algorithm is highlighted in Table 1. For the results that follow it was requested to the system to reconstruct both prior scenarios—six times each—with stopping criteria defined at 0.1% (accordingly to Section 3.6) and 40 mm OctoMap resolution. To accomplish this, 150 poses were sampled by exploration iteration. Table 1 additionally summarizes the statistical information for each scenario, either the requested number of iterations and also the agglomerated distance between the visited poses.

It is evident that when the exploration volume diminishes, fewer iterations, and therefore poses, are required to fully reconstruct the environment. This leads to a shorter travelled distance by the camera’s depth optical frame, but even so, the dispersion of the travelled distance between poses remains approximately the same. This means that the algorithm diversifies its poses no matter the volume to explore, which makes sense considering that this behavior tends to maximize the gain of information by consecutive iterations. Through measuring both volumes a NBV is expected to reveal and the amount that is actually revealed we can analyze not only if the trend expected is confirmed, but also the precision and accuracy of the computed values.

The first two presented plots (Figure 10a,b) are all in terms of absolute values, i.e., in each iteration we break down both amounts of each volume.

Examining Figure 10a it is clear a discrepancy in the first iteration (in the order of half cubic meter), favoring a higher expected volume in comparison to those actually seen. This was predictable, as already discussed, because of the nonexistence of occlusions information before the first pose is reached. After the initial measure, occlusions star becoming more predictable, hence, the discrepancy comes down to just a couple cubic decimeters, coming to less than that in the following NBVs.

In the case of the occluded chair scenario, the revealed volumes in the first iteration are wildly spread and considerably lower than expected (Figure 10b). This is a result of the perturbation in the environment that is the obstacle, promoting big occlusions. The first view always expects to expose large portions of the exploration volume, but then it finds the opaque obstacle that does not allow for further measurements behind it. This contrast is what causes the discrepancy reported by both plots.

Nevertheless, this perturbation is then corrected. Having information about the obstacle, the pose evaluation procedure can take it into account, providing values closer to reality, proving the adaptability for different scenarios.

Notice that the above plots are all monotonously decreasing. This is because the amount of knowledge about a scene, assuming it does not change, can only stay the same or increase. Given that the camera always moves to a pose where it is expected to unveil some volume and, on the path to reach it, is also evaluating the scene, it never stays the same and always increases. This statement is no longer true if we are studying how is the evolution of the volume uncovered by the NBV relative to the total amount of yet unknown—at that point in time—volume present in the scene. Equally to what happens when analyzing the absolute values, the first iteration always gives less information than expected. However, this is the only iteration in which this happens, as posterior poses tend to reveal a bigger fraction of the unknown space than predicted. Figure 10c). Taking into consideration that when moving, the camera is still gathering information about the scene, setting a known state to voxels along the path, explains why this happens since the evaluation algorithm does not account for the information gain along the trajectory. Yet, this ratio tends to get lower the closer the exploration is to finish because, although there is less volume to unveil, that volume is also more scattered, being harder for a single pose to view all of it.

Supporting the stated about the discrepant values in all first iterations, we can see that the objective of favoring big occlusions in the occluded chair scenario was achieved. For this scene, all six explorations predicted, for their first NBV, the reconstruction of 80% to 90% (see Figure 10d), but none even surpassed the 70%, with the lowest barely unveiling 20% of the exploration volume. The discrepancy reached its peak on exploration 6, achieving a difference of over 65%. The dispersion of the measured values (Figure 10d) is also considerable, compared to the previous scenario (Figure 10c), which was expected, given the complete randomness of the FOV, corresponding to the NBV pose, being mostly blocked or not by the obstacle.

Given that the path planning computation is not controlled by the autonomous exploration algorithm (and may be different for the same two poses due to the random nature of the used path planner), the path itself can or cannot be advantageous for an iteration. In the case of exploration 3, it ended up being disadvantageous to the point it required one extra iteration.

The average values only tell part of the story, since our goal is to prove that the algorithm, effectively, predicts with accuracy the least volume of a pose is capable of providing to the system. For that, Figure 11 condenses all autonomous explorations data—that had information about occlusions—correlating the volumes that are expected to be, and actually, unveiled. This means that we are only evaluating iterations that had information to predict occlusions, effectively removing the first iterations, hence ensuring that we are only comparing the robustness of our occlusion prediction algorithm. In this plot we see that a good correlation exists between the data, $R^{2}=0.9543$, seaming to prove the reliability of such predictions.

The correlation line has a lower slope than the ideal, seaming to prove the trend of expecting to unveil a smaller volume than actually happens, due to those voxels that are evaluated while reaching for the NBV.

Analyzing the time consumption per iteration (in a system with an Intel Core i7 4700HQ@2.70 GHz, 8 Gb of RAM and on-board graphics) we see in Figure 12a,b ((Test Scenario 1 and Test Scenario 2) a stabilization around the ten seconds mark after the second iteration. Furthermore, a decreasing trend is, as expected, evident which is consistent with the information provided in Figure 10, which shows a decrease in the number of unknown voxels over time.

Figure 13 shows the size of the scene representation over time. Both test scenarios are analyzed. Results show that the memory size increases over time, which is expected since the exploration of the scene gathers novel information which must be encoded in memory. However, the interesting part is that the increase in memory size does not explode and, even at the end of the experiments, is still manageable. Using other representations such as voxel grids would not be possible for the same voxel resolution.

Case Study 1 allowed proof of the reliability of the system, capable of several autonomous explorations we are confident that the system will, at least, reconstruct the predicted volume and will become more effective. Its high adaptability to different scenarios makes it applicable to various use cases, from an industrial bin-picking process, like the reconstruction of a newly arrived box with scattered pieces for bin-picking, to the modelling of a physical prototype, recreating it on a digital context, using our system to choose the best poses and the actual model being built with point clouds, triangle meshes, or any other desired method.

Both these examples may require different hardware, namely a different manipulator, so the autonomous system must maintain its characteristics in these conditions. To evaluate its performance, in a more industrial scenario, we created a new test scenario, described in the following subsection, to generate a new challenge that permits taking conclusions regarding this need.

### 4.4. Test Scenario 3: Industrial Application

On the third scenario, the goal was to evaluate the portability of the developed system to different hardware (manipulator and camera), and promote its integration with a digital manufacturing stack for interoperability with industrial applications. This effort was integrated on the scope of the H2020 FASTEN project (http://www.fastenmanufacturing.eu/ (accessed on 17 April 2020), where collaborative mobile manipulators are being developed and deployed for the automation of intralogistics operations in cross-sectorial industrial use cases. Within the FASTEN R&D initiative, the Open Scalable Production System (OSPS) is being developed as a solution for the overall interoperability of CPPS in manufacturing scenarios [36]. The OSPS promotes connectivity through an Internet of Things (IoT)-based platform to monitor and coordinate collaborative robotic systems to perform machine tending and pick and place operations in logistics warehouses. Furthermore, the OSPS promotes the development of reusable robot applications through a skill-based robot programming methodology combined with the added support of a digital twin representation of the environment [37].

The proposed system was integrated in the Embraer use case of the FASTEN project, where a mobile manipulator is being developed for interfacing with an automated warehouse for the collaborative assembly of kits of aircraft wings components that are then to be transported to the production lines [34]. This use case aims to enable flexible automation of the parts kit assembly operation by the integration of IoT data streams from the collaborative robotic system with data from the corporate systems, monitoring the operation and coordinating the robot while dealing with a high diversity of parts, dynamism of orders, and unstructured environment. Due to these constraints, high levels of adaptability and flexibility are required from the resulting autonomous system, as to cope with the dynamism and unstructured nature of the organization of containers and parts stored in the automated warehouse, and to promote collaboration and adaptability to co-work and share the space with human operators.

As a result, the proposed autonomous exploration system was employed to allow the collaborative robot to have a better sense of its operational environment, and thus, increase the efficiency and reliability of the bin-picking pick and place operation, currently implemented as a set of robotic skills [36]. As to enable the effective integration of the proposed system with the OSPS, it was necessary to migrate the developed system to be compliant with the definition of a robotic skill which is based on the ROS Action paradigm. Thus, to enable this integration and the usage of the autonomous exploration system within the bin-picking task, the developed methodology was ported to a ROS Action server and corresponding adaptations were done on the orchestration-level ROS Action client.

This implementation also proved the capability of working without MoveIt, since this Application Programming Interface (API) is not used on this system. Instead, this approach relies on robotic skills for manipulator movement that directly interfaced with manipulator controllers. By changing how poses are sampled to, essentially, define a spherical volume where poses can randomly be generated and publishing the NBV to the TF tree, then sending to the robot which frame is its goal pose were enough changes to make it compliant with this approach.

In what regards the hardware changes, and focusing specifically on the manipulator, an Universal Robot’s UR10 was used, which was assembled on top of a moving platform, especially designed for bin-picking operations, composing the mobile manipulator system previously referred. When compared to the previous test scenarios, the UR10 has the advantage of a longer reach, up to 1.3
m, but lacks a spherical wrist, i.e., the last three rotation axis does not intersect in one single point, possibly creating some difficulties on more challenging orientations.

Figure 14 shows the scenario to be explored and Figure 15 demonstrates the process of exploration described in this test scenario.

Driven from the smaller exploration volume ( 1.132
m${}^{3}$), compared to both previous scenarios, the reconstruction task required significantly fewer iterations, three in most cases. The resolution was also higher, at 50 mm, sampling 50 poses, considering that the objective was not to evaluate performance, but rather portability.

This test scenario proved that the developed algorithms can easily be used with other platforms, in its integrity or its constituent parts, for example, the view-sampling and evaluation algorithms. The architecture can also work in harmony with other planners, if there is the need to send pose request to the manipulator, without MoveIt. Moreover, the methodology was ported to a reusable and hardware-agnostic skill-based approach, thus promoting its potential integration in other platforms, environments, and use cases.

### 4.5. Comparison between Automatic and Interactive Explorations

To compare the performance of the developed algorithm against human intelligence, the same task was given to both. This task was performed by six humans and six times by the robot. The scenario was composed of only the shelf of the first scenario in Case Study 1 (see Figure 16a). To both sets—humans and robots—the goal was to reconstruct the given scenario up to 90%. This means that the criteria to terminate the reconstruction is to achieve a remaining unknown volume of less than 10% of the total exploration volume. Additionally, the users had a maximum of eight iterations—four times the best result achieved by the robot—per attempt.

The robot was allowed to use all the tools described in the previous chapters but had only one attempt. On the other hand, humans had two attempts, but the set of tools that they could use was different in each. In the first execution, the human could only see the bounding box that defined the exploration volume (blue lines of Figure 16b) and the colored point cloud captured by the sensor at that moment (see Figure 16b), which from now on will be called the point cloud view. In the second attempt, the bounding box was still visible but instead of the point cloud, the human was allowed to observe the OcTree map being reconstructed (Figure 16c,d) exactly as used by the robot on its procedures, known as volumetric view. The difference was that the human could only evaluate if a pose were the best one or not based on what was its visual perception.

To make the test as fair as possible, the starting pose of the manipulator and its speed during the process were maintained constant. Because it acquires information during the movement, the interactive control was performed with an interactive marker in Rviz (a MoveIt functionality), and we refer to the task as completed when the 10% threshold was reached.

On average, when successfully completing the task, it takes humans a total of 6.20 iterations, using the point cloud view, lowering to 5.00 with the volumetric feedback. These values are almost double the average robot attempt, fixed at only three iterations. Plotting the average value of volume unknown (relative to the total exploration volume) in a given iteration results in Figure 17, where the autonomous system on the robot clearly minimizes—compared to the subjects—the number of movements requested to achieve the goal.

On average each subject requested one less iteration for the task conclusion when having the volumetric view, which tends to indicate that even without the autonomous exploration algorithm, the tools developed help for a faster reconstruction. Further profs for this are more explicit when reviewing the attempts of the individuals that did not meet, in at least one of the executions, the requested amount of volume before the eighth iteration. Table 2 supports this statement, proving the evolution of individuals 6 and 7. With the volumetric set of tools, subject 6 could successfully finish the task, reconstructing more 5.7% of the shelf scene in three fewer movements, an improvement of 7.6% per iteration. On the other hand, subject 7 still did not complete the task, yet only changing the visualization artefact, this subject analyzed more 35.1% volume, translating to a 4.39% increase per iteration.

Another key aspect that is interesting to compare is the tendencies of movements chosen. Two measures were taken: the Euclidean distance and the rotation (based on the axis-angle representation) between two consecutive poses. Please note that the frustum section is not square, it is actually a rectangle, making the roll angle of the camera influence the number of voxels sensed, as also does the distance to the volume. Table 3 seems to indicate a preference by the robot to select poses that are further away and demand more from the rotational component, respectively 1.5 and 1.4 times the average human, indicating that the subjects did not account for the frustum geometry as well as the autonomous algorithm did. Yet, further validation is still necessary due to the limited amount of data and analyses.

The data collected during testing indicates that the autonomous system is, in fact, able to explore the volume in a more efficient manner than humans. One thing that is not measurable but occurred frequently during testing was the choosing of poses (by the algorithm) that were not intuitive at first glance, nonetheless, they were quite profitable—very frequently gave the highest score of the sampled set—and made sense when gathering together all the discussed factors.

The developed and improved ROS packages regarding this work are publicly available:SmObEx—Smart Object Exploration (https://github.com/lardemua/SmObEx)FANUC M6iB/6S Support and MoveIt Config (https://github.com/ros-industrial/fanuc/pull/264)OctoMap Tools (https://github.com/miguelriemoliveira/octomap_tools)

Also, a list of videos of the working system is accessible through a playlist (https://www.youtube.com/playlist?list=PLlFdlPPJjdmmCzNEbp_u2mB5nFM9WHmFo (accessed on 1 July 2019)).

## 5. Conclusions

The main goal of this work was to provide a robotic manipulator with autonomous exploration capabilities, with or without prior knowledge of the scene. For this, a six Degrees of Freedom (DoF) manipulator was used with an RGB-D camera mounted on the end-effector link. To the best of our knowledge, this is the first approach that tackles the NBV problem for a robotic manipulator in real-world scenarios.

Several algorithms were developed to match the requirements of this work. For example, OctoMap’s implementation of an updatable OcTree structure does not explicitly store which voxels are unknown. Tackling this issue requested the building of a parallel OcTree, generated based on the implicit information extracted from the OctoMap representation. This process is not perfect since it always requests both structures and it is necessary to keep in mind that the parallel OcTree has voxels marked as free that are corresponding to unknown in the original one. Yet, considering the lack of tools for this procedure, the algorithm worked as intended given the fact that it was possible to successfully and reliably use this second OcTree to assess which voxels had to be visited. Furthermore, an exploration algorithm was developed to solve the NBV problem. These algorithms can also work separately, making them useful for other applications domains.

The full methodology was integrated and tested in an industrial prototype for intralogistics operations on the scope of a European-funded R&D initiative, thus abiding by the potential industrial application of the developed solution. It was demonstrated that the proposed methodology proved to be reliable and stable. The exploration performance of the proposed approach was also compared with that of human users. In these experiments, the automatic exploration system was able to reconstruct the scenario in half the iterations required, on average, by the humans.

This work could act as a foundation for several other research efforts. The NBV selecting algorithm could be used in a more industrial oriented application, as is bin-picking. In this kind of task, removing one piece can help unveil others, which then becomes more easily accessible. Using the developed system for pose selection, the manipulator could force the passage through strategically placed waypoints that would maximize its knowledge about the constantly changing scene [38]. One other branch from this work could be a vehicle capable of exploring any ambient, by mounting the manipulator in a moving platform. With a path planner developed to integrate both platform and manipulator, the pose generation would not be bounded by the manipulator’s reach by but instead only by the volume requested for the exploration. The platform plus manipulator would be able to move to the selected NBV, avoiding obstacles in the way. In this scenario, a fleet of vehicles could all contribute to the same exploration, requiring collaborative behaviors for coordination and information sharing between them [39] or even to offload the most computationally intensive tasks to an external infrastructure (cloud robotics) [40]. Lastly, possible combining the RGB data, the robot could detect a new object placed before it, to then classify it [41]. This classification would be useful for different tasks, for example, choosing the proper gripper configuration to grab it or to place it in the desired container, if we think of a task where the goal of the robot is to separate the objects by categories.

Bearing everything in mind, the contributions done by this work have a large potential for several industrial applications, but also for prototype development, being considered by the authors as a good starting point in the path to provide manipulators with awareness about their surroundings.

## Figures and Tables

**Figure 1 sensors-20-04331-f001:**
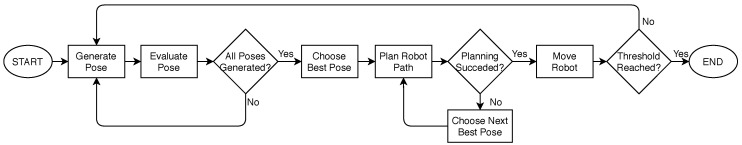
Flowchart of the full exploration algorithm.

**Figure 2 sensors-20-04331-f002:**
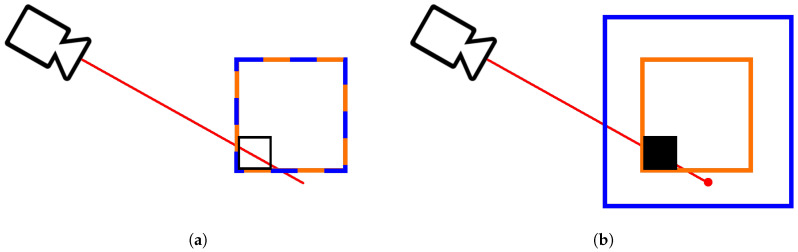
Exploration volume vs. point cloud filter volume: (**a**) coincident and since the ray is reflected outside the point cloud filter volume, it is impossible to know in which state the black voxel is: (**b**) point cloud filter volume larger than exploration volume, the point is registered and the state of the voxel is known.

**Figure 3 sensors-20-04331-f003:**
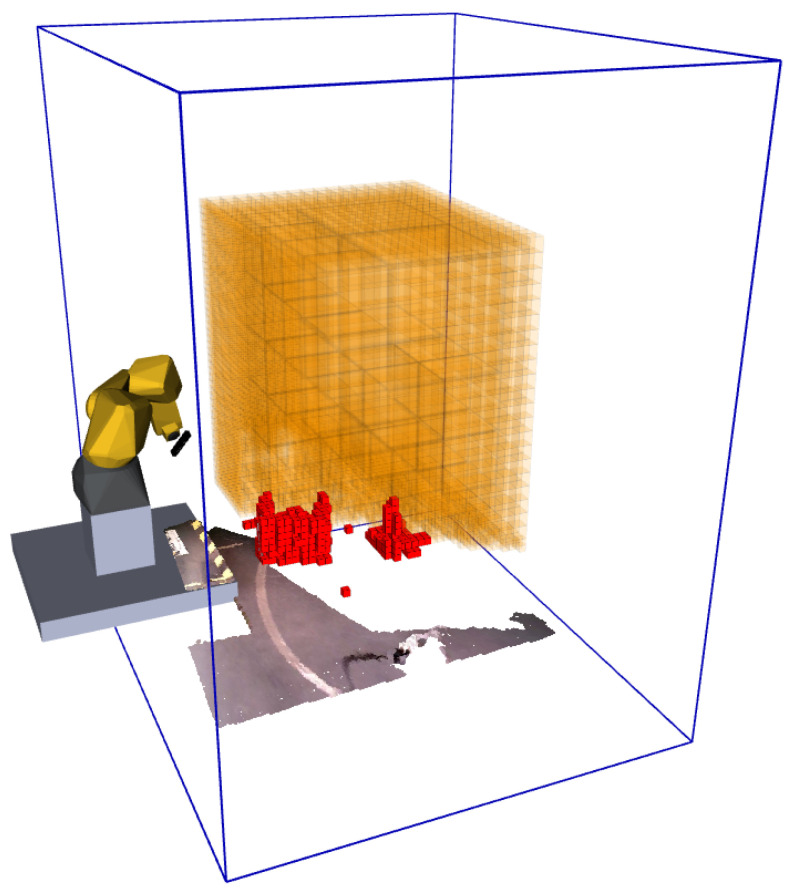
Definition of the point cloud filter volume (each side is two times bigger than those of the exploration volume). Exploration volume (orange represents unknown space, red occupied space. Free space hidden for visualization purposes). Blue lines mark the boundaries of the point cloud filter volume. Filtered point cloud also visible.

**Figure 4 sensors-20-04331-f004:**
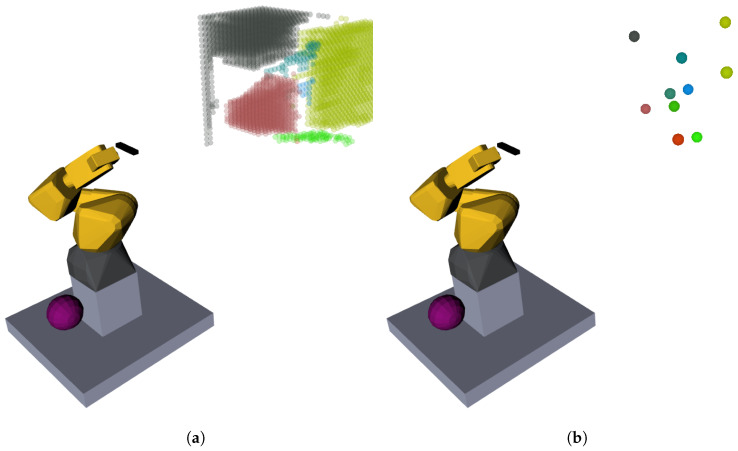
Example of ten clusters found after an exploration iteration. (**a**) Point clouds (points are the center points of the unknown voxels) that define each cluster. (**b**) Respective clusters centroids. Different colors represent different clusters.

**Figure 5 sensors-20-04331-f005:**
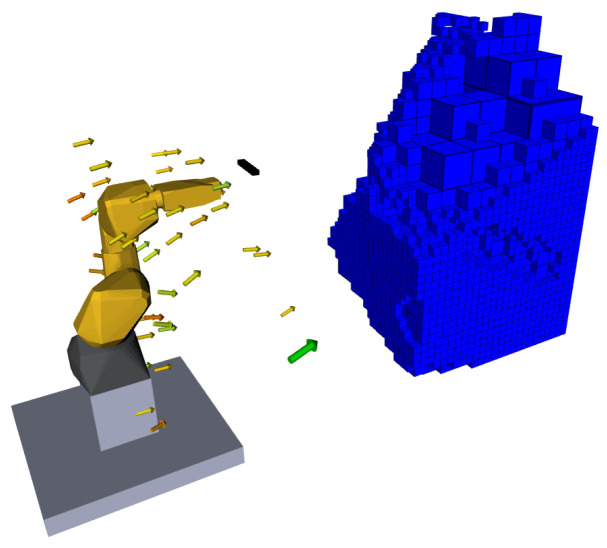
Example of 50 poses generated using a conditional randomized approach, bounded by the systems reach. In this case, there is a single unknown voxel cluster, to which all sampled poses are pointing at. Unknown space is hidden for better visualization. Voxels in blue are the ones expected to be evaluated based on the best pose of those sampled, colored as green.

**Figure 6 sensors-20-04331-f006:**
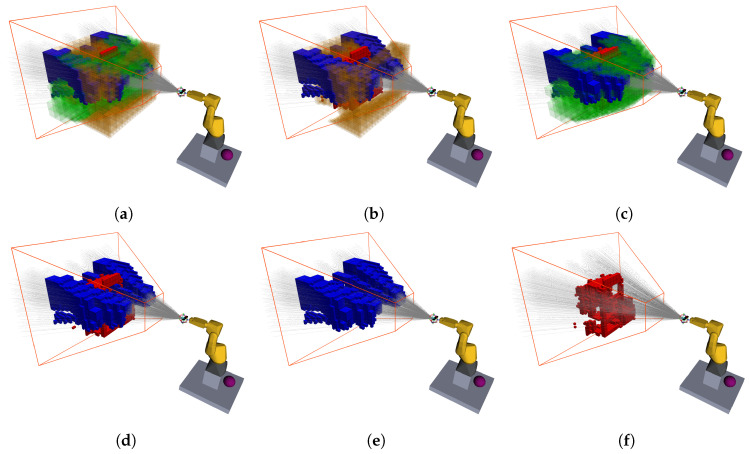
Evaluated pose. This figure shows the beams passing the free space (green) and stopping when they hit an occupied voxel (red). The voxels that potentially will be known are marked in blue. In (**a**), everything is shown. In (**b**), it is visible the unknown, occupied, and expected to be known volumes. (**c**) is similar to (**b**), but the unknown volume was removed and the free added. In (**d**), only the occupied and expected to be known volumes are showing. In (**e**,**f**), the expected to be known and occupied volumes are shown.

**Figure 7 sensors-20-04331-f007:**
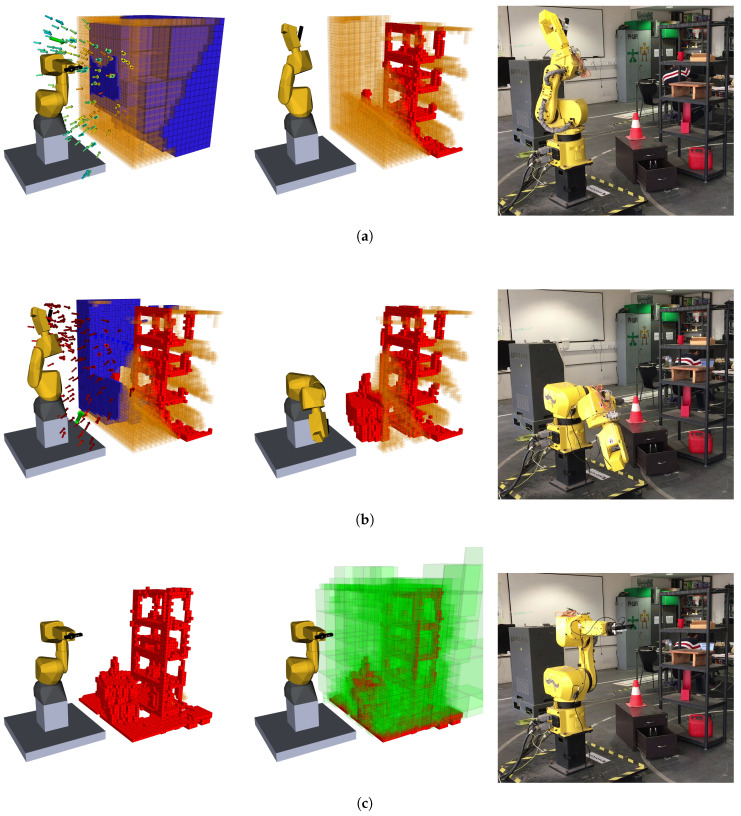
A sequence of iterations for the complete reconstruction of the shelf and cabinet scene. On the left are all the possible poses (the more blue the better score, and the redder the worst), the chosen NBV (in green and bigger), the unknown voxels in orange and the voxels expected to be known on blue. In the middle is what the pose actually improves the model, red voxels are occupied, green represents the free space. On the right is the actual pose of the robot. (**a**) shows the first exploration iteration, (**b**) the reconstruction improvements of an intermediate iteration and (**c**) is the result when the process is concluded.

**Figure 8 sensors-20-04331-f008:**
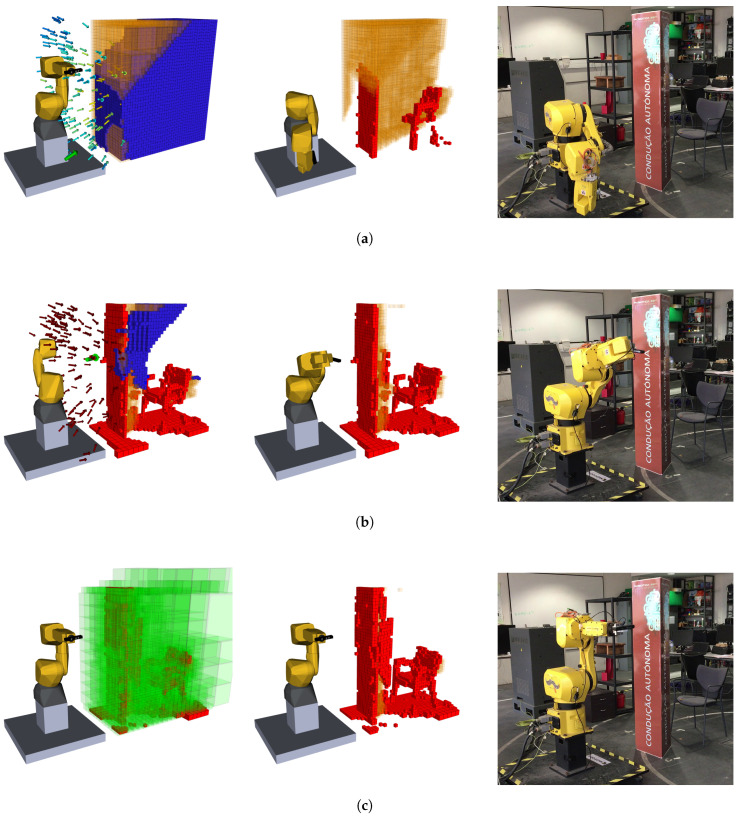
A sequence of iteration for the complete reconstruction of the occluded chair scenario. On the left are all the possible poses (the more blue the better score, and the redder the worst), the chosen NBV (in green and bigger), the unknown voxels in orange and the voxels expected to be known on blue. In the middle is what the pose actually improves the model, red voxels are occupied, green represents the free space. On the right is the actual pose of the robot. (**a**) shows the first exploration iteration, (**b**) the reconstruction improvements of an intermediate iteration and (**c**) is the result when the process is concluded.

**Figure 9 sensors-20-04331-f009:**
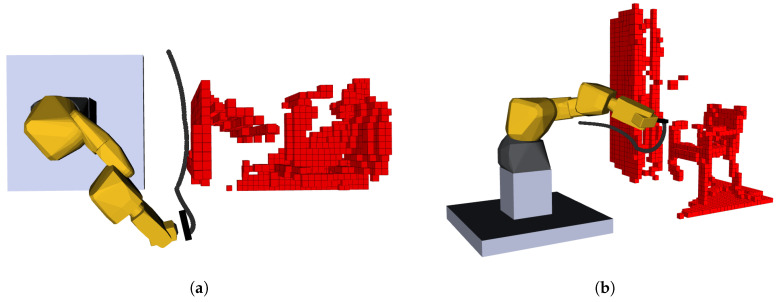
Path taken by the depth optical frame to reach the goal pose, without colliding with the space already known to be occupied. (**a**) is the top view and (**b**) is a perspective view.

**Figure 10 sensors-20-04331-f010:**
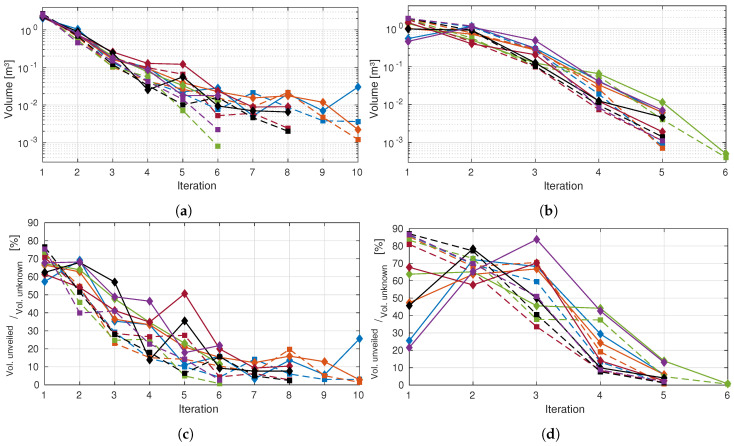
Volumes expected to be unveiled (dashed line) and actually unveiled (full line) over (**a**) the exploration in the shelf and cabinet scenario and (**b**) the exploration in the occluded chair scenario. Fractions of the volume expected to be unveiled (dashed line) and actually unveiled (full line), (**c**) relative to existing unknown volume, over the exploration in the shelf and cabinet scenario and (**d**) relative to existing unknown volume, over the exploration in the occluded chair scenario.

**Figure 11 sensors-20-04331-f011:**
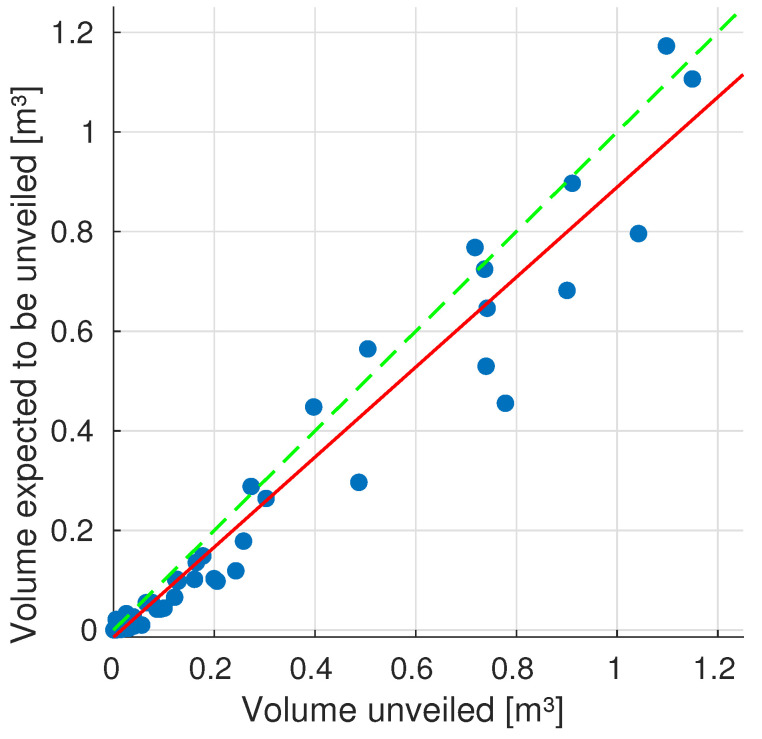
Scatter plot (for both scenarios) correlating in all iterations, except the first one, the expected and unveiled volumes. The green dashed line represents where the data should lay if there was an ideal correlation. The red line is the actual correlation between what is expected and what is actually measured, with $R^{2}=0.9543$.

**Figure 12 sensors-20-04331-f012:**
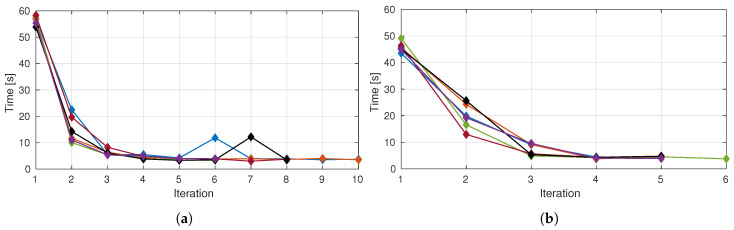
Time consumed, in each iteration, to choose the NBV, for (**a**) Test Scenario 1 and (**b**) Test Scenario 2.

**Figure 13 sensors-20-04331-f013:**
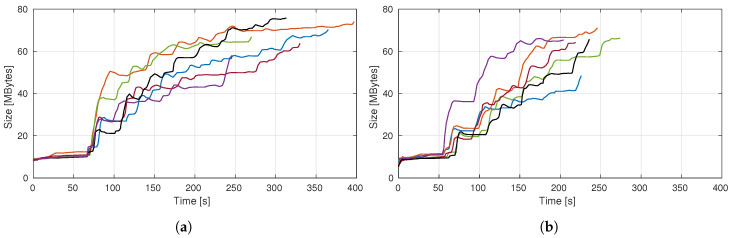
Size of scene representation over time for several experiments: (**a**) Test Scenario 1 (**b**) Test Scenario 2.

**Figure 14 sensors-20-04331-f014:**
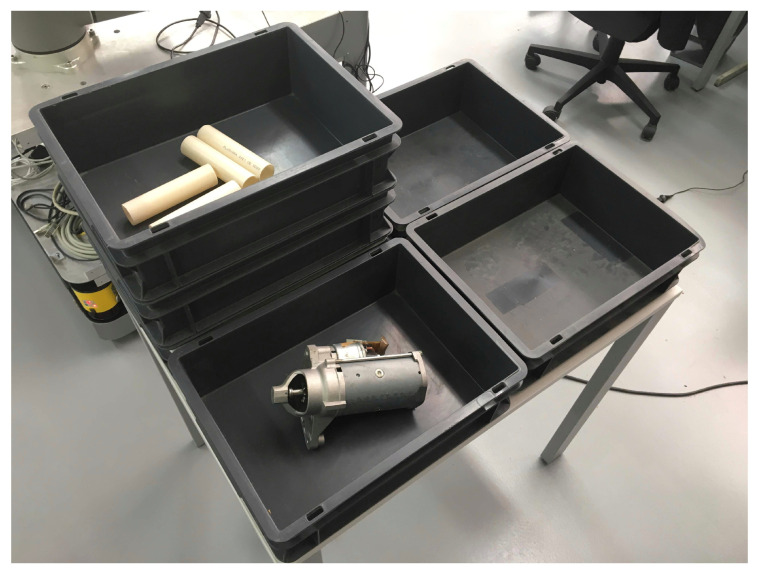
Set of boxes and their contents used for this case study. In the upper left corner box, there are four plastic tubes. In the lower-left box, it is visible a large-scale component. This assembly was considered to be particularly relevant and similar to potential scenarios found at the FASTEN’s Embraer use case plant in Évora, Portugal.

**Figure 15 sensors-20-04331-f015:**
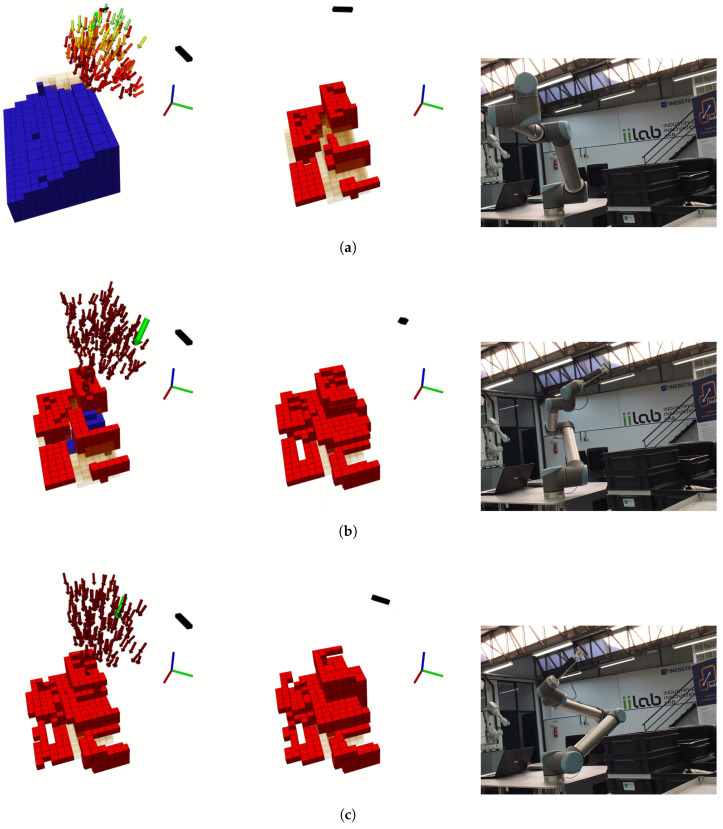
A sequence of iterations for the complete reconstruction of the scene of the third test scenario. On the left are all the possible poses (the more blue the better score, and the redder the worst), the chosen NBV (in green and bigger), the unknown voxels in orange and the voxels expected to be known on blue. In the middle is what the pose actually improves the model, red voxels are occupied, green represents the free space. On the left is the actual pose of the robot in the industrial scenario. The axis is the correspondent to the base link of the manipulator. (**a**) shows the first exploration iteration, (**b**) the reconstruction improvements of an intermediate iteration and (**c**) is the result when the process is concluded.

**Figure 16 sensors-20-04331-f016:**
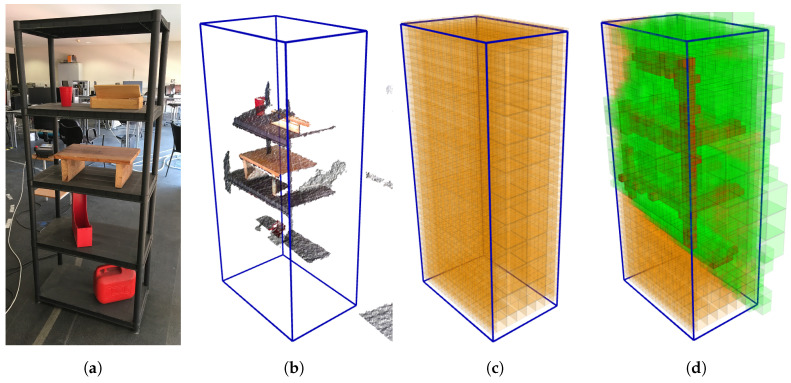
Visualization examples of the scenario (**a**) and tools given to humans to explore the volume. The tools were the bounding box defining the exploration volume with the colored point cloud (**b**), the unknown voxels (**c**), and the reconstruction OcTree (**d**).

**Figure 17 sensors-20-04331-f017:**
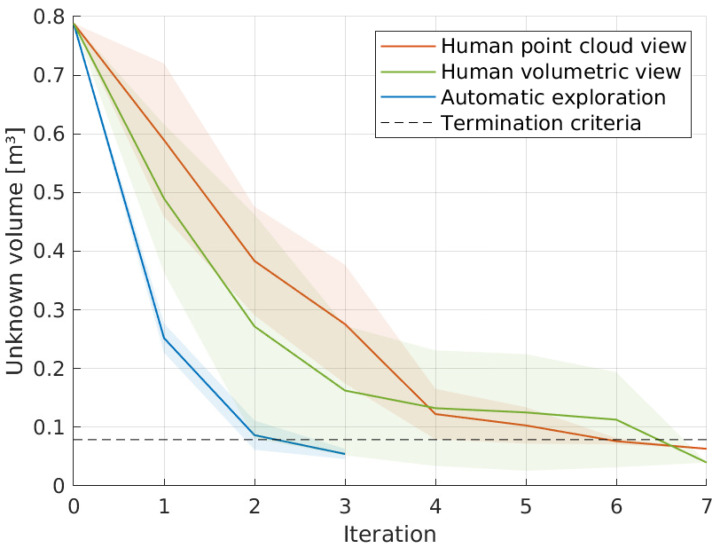
Average fraction of volume still unknown in each iteration and corresponding standard deviation relative to it. The dashed line is the cut off value.

**Table 1 sensors-20-04331-t001:** Statistical results of the explorations in Case Study 1, shelf and cabinet and occluded chair scenarios.

Scenario	Exploration Volume [m³]	Number of Iterations		Total Distance [m]	
		Mean	Standard Deviation	Mean	Standard Deviation
Shelf and Cabinet	3.531	8.00	1.789	8.525	1.084
Occluded Chair	2.140	5.17	0.408	6.365	1.089

**Table 2 sensors-20-04331-t002:** Comparison between using the point cloud view and the volumetric view for subjects that did not achieve the goal in the maximum number of iterations. In bold are the subjects that did not achieve the goal.

Subject	Point Cloud View		Volumetric View	
	Iterations	% Volume Unknown	Iterations	% Volume Unknown
1	5	7.6	4	2.5
2	6	9.8	7	5.0
3	6	8.9	5	6.7
4	4	7.8	6	7.0
5	7	8.0	3	4.5
**6**	**8**	**13.6**	**5**	**7.9**
**7**	**8**	**50.6**	**8**	**15.5**

**Table 3 sensors-20-04331-t003:** Comparison, per iteration, of the mean translation and rotation of the camera’s depth optical frame, between the robot and human behaviors.

	Robot		Human	
	Mean	Standard Deviation	Mean	Standard Deviation
Distance [m/iteration]	1.240	0.136	0.840	0.317
Rotation [°/iteration]	127.633	22.633	91.910	28.906

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
