# Peer review of "Autonomous Scene Exploration for Robotics: A Conditional Random View-Sampling and Evaluation Using a Voxel-Sorting Mechanism for Efficient Ray Casting"

_sensors, 2020, doi:10.3390/s20154331_

Round 1

Reviewer 1 Report

Good job, valuable paper, however, it is worth introducing some minor improvements as indicated below:

  • In sections "1. Introduction" and "2. State of the Art" it is worth adding more references to (existing) items in the list of literature to fully convincingly substantiate the facts given there. In particular, this note refers to "2. State of the Art".
  • Although they seem quite obvious, it would be worth justifying equations (1) and (2) or specifying sources as citation.
  • Footnotes 2 to 6 should be completed by entering the date of access.

Author Response

“In sections "1. Introduction" and "2. State of the Art" it is worth adding more references to (existing) items in the list of literature to fully convincingly substantiate the facts given there. In particular, this note refers to "2. State of the Art".”

First we would like to thank the reviewer for his careful review. We agree with the reviewer that additional background was useful. To address this, several references were added to “2. State of the Art”. The references are listed below:

To back what we introduce as “The ability to explore a new environment, quickly and efficiently, developed to increase our survival skills, is intrinsic to humans” we now reference  {wu_generalization_2018}. This proposed work goes through the aspects of human behaviour that can be traced to a “(...) type of search problem (...)”.

To support the relatively good quality of inexpensive RGB-D sensors, we referenced {rausher_comparison_2016} in which a subset of this kind of devices are compared against some other, more capable, devices. They found that “The structured light sensors Microsoft Kinect and ASUS Xtion Pro Live are very accurate for close ranges up to 3.5 m, where the increase of noise can be assumed to be linear.”

In the works of {jiang_simultaneous_2019} and {gu_rapid_2008} elevation maps are described and their possible implementations are tested. We think that adding these references can be helpful for a reader that may require more information about the topic. 

The methodology behind raycasting and other applications other than the generation of  point clouds, is studied by {sauze_raycast_2010}. In this publication, raycasting is analysed in the scope of collision avoidance for sailing robots. Authors analyse the benefits, procedures and applicability of this method for that particular application, while describing in detail the raycasting methodologies.

The introduction to voxel grids is backed by {kong_point_2019}.This work starts from a voxel grid and tries to obtain a point cloud suitable to the object which the voxel grid describes. For this, a detailed description of voxel grids is provided.

With regards to the efficiency of voxel grids, {elseberg_one_2013} proposes an algorithm to improve the efficiency of OcTree’s in the case of 3D laser scans. The efficiency of current implementations is discussed and an explanation of the MURIEL method proposed by {konolige_improved_1997} was added.

Finally, efficient implementations of OcTrees are debated and analised on {elseberg_efficient_2011}, {canelhas_survey_2018} and {han_towards_2018}, which are also cited.

“Although they seem quite obvious, it would be worth justifying equations (1) and (2) or specifying sources as citation.”

As per the reviewer’s request we added a justification for equation (1): “The advantage of this method is that the total number of poses is independent of the amount of clusters found, allowing for a more stable time consumption per exploration iteration, when compared to the amount of poses that is set for each found cluster.”

Regarding equation (2) it is discussed in lines 457 to 462, which state “To rank all the poses, a term of comparison has to be set. In this work, when estimating how many voxels a pose will evaluate, we expressed this value as a volume. Then, it becomes straightforward that the term of comparison should also be a volume. As referred in algorithm 1, the order in which the voxels are intersected is a useful information. The first voxel which is intersected is always assessed. As other voxels which lie behind the first (along the casted ray) will be visible or not depending on whether an occupied voxel surges in front of it.”

“Footnotes 2 to 6 should be completed by entering the date of access.”

The footnotes were edited and now contained the date of access.

Reviewer 2 Report

Although the paper presents some experimental evidence, unfortunately this reviewer does not perceive a theoretical foundation that demonstrates scientific advances, but rather an integration-work on: frontier-void-based techniques, active volumetric 3D reconstruction, and optimal motion planning and stopping. Although integration has its own merit, artificial/novel strategies for this purpose should be proposed.

It is also recommended adding detailed and profuse statistics on performance, cost, and intervals, as well as location and orientation errors.

Some other form-comments are:

Introduction should be more concise and with more references to related-research.
Some phrases (some too long) must be connected/verified and many typos corrected.
Preferably, use a very formal and anti-personal language.
Selecting a 5X point cloud filter volume, respect to the exploration volume, must be justified.
In Figure 4b, it is difficult for the reader to visually locate the centroid.

Author Response

“Although the paper presents some experimental evidence, unfortunately this reviewer does not perceive a theoretical foundation that demonstrates scientific advances, but rather an integration-work on: frontier-void-based techniques, active volumetric 3D reconstruction, and optimal motion planning and stopping. Although integration has its own merit, artificial/novel strategies for this purpose should be proposed.”

We thank the reviewer for his review of the manuscript. We do agree that our work was to a large extent an integration work. But we would like to highlight that we do propose several novelties w.r.t the state of the art, in particular the manuscript “(...) proposed a viewpoint evaluation methodology which sorts the voxels for which rays are cast, which enables the algorithm to skip a great number of tests and run very efficiently, thus allowing for more viewpoints to be tested for the same available time;”

Furthermore, this manuscript provides detailed results of an automatic exploration system, in addition to an uncommon comparison against the exploration performance of human subjects.

Finally, we also agree with the reviewer when he/she merits the integration work, in particular since we used advanced software tools and propose to release the code to the community.

“It is also recommended adding detailed and profuse statistics on performance, cost, and intervals, as well as location and orientation errors.”

We have tried to provide detailed results, describing several scenarios and experiments (see, for example, Table 1). In addition to that, we also provided a comparison with the exploration abilities of human subjects. Nonetheless, we agree with the reviewer in that additional statistics improve the manuscript. Thus, we have added as per the reviewers request 4 new graphs which characterize the performance of the system. The first pair of graphs show the time per iteration in the first two test scenarios (Figure 12). The second pair of graphs detail the size of the memory required to store the scene representation and how this evolves as the exploration progresses (Figure 13).

With these additions we believe that the experiments and performance of the proposed system are more detailed.

“Introduction should be more concise and with more references to related-research.”

We have revised the introduction and made an effort to make it more concise. We have added several new references, as pointed out in the answer to the first question made by  Reviewer 1.

“Some phrases (some too long) must be connected/verified and many typos corrected. (...) Preferably, use a very formal and anti-personal language.”

We have revised the complete manuscript in an attempt to remove long sentences and use more formal language.

“Selecting a 5X point cloud filter volume, respect to the exploration volume, must be justified.”

We thank the reviewer for his/her comment. We have added a justification as follows: “In these experiments, a region five times larger than the exploration volume was used. This value was empirically obtained as the best compromise between memory usage and the performance of the map reconstruction.”

“In Figure 4b, it is difficult for the reader to visually locate the centroid.”.

We thank the reviewer for his/her comment. The colors were intended to help the reader connect the cluster with the respective centroids. Having said that, we increased the size of both images and removed excessive white borders to help the reader to better locate de centroids.

Reviewer 3 Report

In this paper, the authors  presented lots of work on developing tools for a robotic system to be able to explore a scene autonomously. In order to estimate the volume that maybe observed, The authors proposed an algorithm, which is based on ray casting defined according to the RGB-D sensor’s characteristics. while the RGB-D camera is mounted on the end effector of a 6-dof robot. This paper has some good reference. The reviewer suggests it to be accepted.

However, there are still some points should be taken in to consideration.

1. How to choose the keyword "ROS"? The reviewer cannot see ROS in the title or abstract. Generally the keyword should be selected from the title or the abstract.

2. In section 2 "State of the Art", there are FIVE pages to illustrate such "state of art". From the reviewer's points, the authors should give a different style to illustrate such problem with some figures or charts. And then the readers can read easily.

3. The appearance sequence of Figure 2 and Figure 3 should be noted.

4. The conclusion should be rewritten. The authors just need to show the important conclusions.

Author Response

“In this paper, the authors  presented lots of work on developing tools for a robotic system to be able to explore a scene autonomously. In order to estimate the volume that may be observed, the authors proposed an algorithm, which is based on ray casting defined according to the RGB-D sensor’s characteristics. while the RGB-D camera is mounted on the end effector of a 6-dof robot. This paper has some good reference. The reviewer suggests it to be accepted.”

We would like to thank the reviewer for his comments.

“1. How to choose the keyword "ROS"? The reviewer cannot see ROS in the title or abstract. Generally the keyword should be selected from the title or the abstract.”

We thank the reviewer for their comment. We agree and have now modified the abstract to include the keyword “ROS”.

“2. In section 2 "State of the Art", there are FIVE pages to illustrate such "state of art". From the reviewer's points, the authors should give a different style to illustrate such problem with some figures or charts. And then the readers can read easily.”

We have entirely revised section 2 and have tried to make it more clear and using a different style. Also, we added several new references (see our answer to the first question 1 of Reviewer 1) to better support our claims. In our opinion this section is now more readable.

“3. The appearance sequence of Figure 2 and Figure 3 should be noted.”

We agree with the reviewer in that we should change the sequence of said Figures. We have corrected the problem.

“4. The conclusion should be rewritten. The authors just need to show the important conclusions.”

We agree with the reviewer and have re-structured Section 5 (Conclusion) to be more straight to the point. It is the understanding of the authors that the revised version of Section 5 focuses more on highlighting the scientific contributions proposed in the manuscript, without repeating previously addressed topics. In addition, the revised Conclusion also highlights future work paths, further attesting to the potential reach of the proposed contribution.

Round 2

Reviewer 2 Report

This reviewer thanks the authors for the revisions made to the article. The presentation quality has improved, from my point of view, enough to be published.

I only recommend indicating in Figure 17 the meaning of each shaded area if it has one.

Furthermore, I suggest adapting the title to the actual scientific contribution (voxel sorting, human exploration comparisons, etc.) and adding the quantitative results obtained to the abstract (perhaps it is convenient to summarize it since it is extensive) emphasizing its contribution.

Author Response

“This reviewer thanks the authors for the revisions made to the article. The presentation quality has improved, from my point of view, enough to be published.”

The manuscript has improved thanks to the insightful comments of all reviewers. Thank you.

“I only recommend indicating in Figure 17 the meaning of each shaded area if it has one.”

We acknowledge the missing of this information throughout the manuscript and thank the reviewer for noticing it. We changed Figure 17 caption to “Average fraction of volume still unknown in each iteration and corresponding standard deviation relative to it. The dashed line is the cut off value.” and think that it is now clear the meaning of the shaded areas.

“Furthermore, I suggest adapting the title to the actual scientific contribution (voxel sorting, human exploration comparisons, etc.)”

We have considered the reviewer's suggestion and have altered the title of the manuscript to the following: “Autonomous Scene Exploration for Robotics: A conditional random view sampling and evaluation using a voxel sorting mechanism for efficient ray casting”. We believe this new title is more suitable to describe the contributions of this work.

“(...) and adding the quantitative results obtained to the abstract (perhaps it is convenient to summarize it since it is extensive) emphasizing its contribution.”

We have rewritten the abstract using the reviewer’s suggestions. However, results are indeed quite extensive and therefore we could not come up with an unique quantitative figure to describe them. We did try to make the abstract more clear now, in particular w.r.t. the results which were achieved.